# Focal Attention for Long-Range Interactions in Vision Transformers

**Jianwei Yang**[1]   **Chunyuan Li**[1]   **Pengchuan Zhang**[1]   **Xiyang Dai**[2]   **Bin Xiao**[2]
**Lu Yuan**[2]   **Jianfeng Gao**[1]
[1]Microsoft Research at Redmond, [2]Microsoft Cloud + AI
{jianwyan,chunyl,penzhan,xidai,bixi,luyuan,jfgao}@microsoft.com

## Abstract

Recently, Vision Transformer and its variants have shown great promise on various computer vision tasks. The ability of capturing local and global visual dependencies through self-attention is the key to its success. However, this also brings challenges due to quadratic computational overhead, especially for the high-resolution vision tasks (*e.g.*, object detection). Many recent works have attempted to reduce the cost and improve model performance by applying either coarse-grained global attention or fine-grained local attention. However, both approaches cripple the modeling power of the original self-attention mechanism of multi-layer Transformers, leading to sub-optimal solutions. In this paper, we present *focal attention*, a new attention mechanism that incorporates both fine-grained local and coarse-grained global interactions. In this new mechanism, each token attends its closest surrounding tokens at fine granularity and the tokens far away at coarse granularity, and thus can capture both short- and long-range visual dependencies efficiently and effectively. With focal attention, we build a new variant of Vision Transformer models, called *Focal Transformers*, which achieve superior performance over the state-of-the-art (SoTA) Vision Transformers on a range of public image classification and object detection benchmarks. In particular, our Focal Transformer models with a moderate size of 51.1M and a large size of 89.8M achieve **83.6%** and **84.0%** Top-1 accuracy, respectively, on ImageNet classification at $224 \times 224$. When employed as the backbones, Focal Transformers achieve consistent and substantial improvements over the current SoTA Swin Transformers [43] across 6 different object detection methods. Our largest Focal Transformer yields **58.7/59.0** box mAPs and **50.9/51.3** mask mAPs on COCO mini-val/test-dev, and **55.4** mIoU on ADE20K for semantic segmentation, creating new SoTA on three of the most challenging computer vision tasks. Our code is available at: `https://github.com/microsoft/Focal-Transformer`.

## 1   Introduction

Nowadays, Transformer [57] has become a prevalent model architecture in natural language processing (NLP) [20, 6]. In the light of its success in NLP, there is an increasing effort on adapting it to computer vision (CV) [47, 50]. Since its promise firstly demonstrated in Vision Transformer (ViT) [21], we have witnessed a flourish of full-Transformer models for image classification [55, 60, 64, 43, 76, 56], object detection [8, 85, 79, 18] and semantic segmentation [58, 62]. Beyond these static image tasks, it has also been applied on various temporal understanding tasks, such as action recognition [40, 78, 10], object tracking [13, 59], scene flow estimation [38].

The self-attention mechanism is arguably the key component that differentiates Transformers from the widely used convolutional neural networks (CNNs) [37] in computer vision. At each Transformer layer, self-attention enables global content-dependent interactions among different image regions for

35th Conference on Neural Information Processing Systems (NeurIPS 2021).

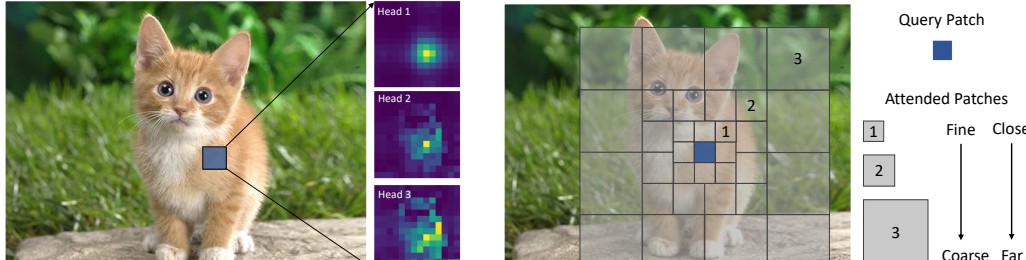

Figure 1: Left: Visualization of the attention maps of the three heads at the given query patch (blue) in the first layer of the DeiT-Tiny model [55]. Right: An illustrative depiction of focal attention mechanism. Three granularity levels are used to compose the attention region for the blue query.

modeling short- and long-range dependencies, respectively. Through the visualization of full self-attention results[1], we indeed observe that self-attention learns to attend local surroundings (like CNNs) and the global contexts at the same time, as illustrated in Fig. 1 (Left). Nevertheless, when dealing with high-resolution vision tasks such as object detection or segmentation, an efficient implementation of a global and fine-grained self-attention becomes non-trivial due to the quadratic computational cost with respect to the number of tokens in feature maps. Recent works have alternatively exploited either a coarse-grained global self-attention [60, 64] or a fine-grained local self-attention [43, 76, 56], for the sake of reducing the computational cost. However, both approaches cripple the power of the original full self-attention *i.e.*, the ability to simultaneously capture local and global visual dependencies.

In this paper, we present a new attention mechanism to capture both short- and long-range interactions in Transformer layers for high-resolution input images. Considering that the visual dependencies between the nearby (local) regions are usually much stronger than the dependencies between the regions that are far away, we perform the fine-grained attention only in local regions while the coarse-grained attention globally. As depicted in Fig. 1 (Right), a query token in the feature map attends its closest local surroundings at the finest granularity as itself. However, when it goes to the regions far away, it attends to *summarized* tokens to capture coarse-grained visual dependencies. We call this new mechanism *focal attention*, as each token attends the others in a focal manner. We will show in this study that focal attention allows to effectively model visual dependencies among all regions covering the whole high-resolution feature maps while introducing much less number of tokens in the computation than that in the standard self-attention mechanism.

Equipped with focal attention, a series of *Focal Transformers* are developed and validated via a comprehensive empirical study across three core vision tasks, including image classification, object detection and segmentation. Results show that Focal Transformers consistently outperform the SoTA Vision Transformers across various settings (i.e., in model sizes and complexities). Notably, the small Focal Transformer with 51.1M parameters achieves 83.6% top-1 accuracy on ImageNet-1K, and the base model with 89.8M parameters obtains 84.0% top-1 accuracy. In the fine-tuning experiments for object detection, Focal Transformers consistently outperform the SoTA Swin Transformers [43] across six popular object detection methods. Our largest Focal Transformer model achieves **59.0** box mAP and **51.3** mask mAP on COCO test-dev for object detection and instance segmentation, respectively, and **55.4** mIoU on ADE20K for semantic segmentation. These results demonstrate that focal attention is highly effective in modeling the global interactions in Vision Transformers.

## 2 Related work

**Vision Transformers**. Vision Transformer (ViT) is first introduced in [21]. It applies a standard Transformer, originally developed for NLP [57], to encode an image by analogously splitting the image into a sequence of visual tokens. It has demonstrated superior performance to CNNs such as ResNet [33] on multiple image classification benchmarks, when trained with sufficient data [21] and carefully designed data augmentation and regularization methods [55]. The results thus inspire researchers to explore the applications of ViT on various vision tasks beyond image classification, such as self-supervised learning [14, 9, 39], object detection [8, 85, 79, 18] and semantic segmentation [58, 62, 81]. There are also increasing number of studies for improving ViT via data-efficient

---

[1]DeiT-Tiny model, checkpoint downloaded from `https://github.com/facebookresearch/deit`.

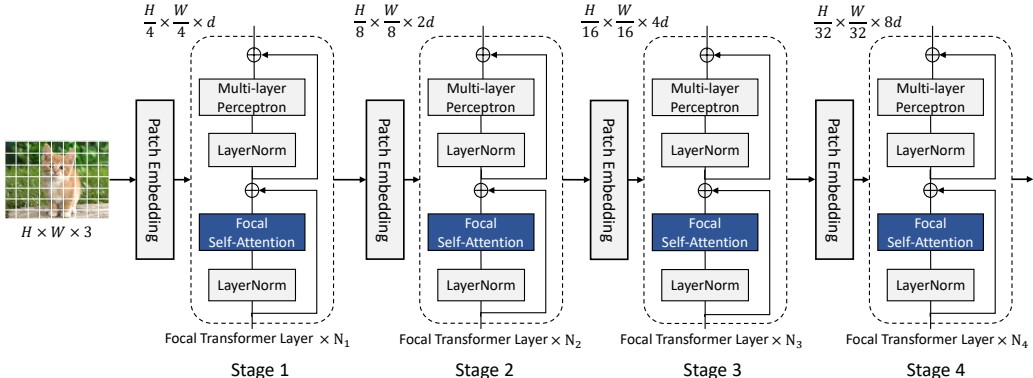

Figure 2: Model architecture for our Focal Transformers. As highlighted in light blue boxes, our main innovation is the proposed focal attention in each Transformer layer.

training [55], improved patch embedding/encoding [16, 71, 31], integrating convolutional projections into transformers [64, 70], and using multi-scale architectures and efficient self-attention mechanisms for high-resolution vision tasks [60, 64, 43, 76, 15]. Recent surveys include [36, 30, 36]. This paper focuses on improving the self-attention mechanism of ViT for encoding high-resolution images.

**Efficient global and local self-attention**. In many real-world tasks, Transformers need to cope with a large number of input tokens, such as long documents in NLP and high-resolution images in computer vision (CV). Recently, many efficient self-attention mechanisms have been proposed to deal with the quadratic computational and memory cost incurred by the standard self-attention mechanism. On one hand, a number of works in both NLP and CV resort to coarse-grained global self-attention (i.e., attending the down-sampled or summarized tokens) to capture the long-range interactions [49, 46, 60, 64, 31, 23]. Although this approach improves the model efficiency, it loses the detailed context information surrounding the query tokens. On the other hand, to make the computational cost manageable, various local fine-grained attention mechanism (i.e., attending neighboring tokens within a pre-set window size) are used for both NLP [3, 74, 1] and CV [56, 43, 76]. In this paper, we argue that both global and local attentions are important for model performance. This is also validated by some recent studies that aim to improve CNNs by incorporating ways of modeling global attentions [35, 63, 61, 68, 2, 7, 51]. The standard self-attention mechanism used by ViT can indeed learned both types of attentions, as shown in Fig. 1 (Left). But it often incurs a prohibitively high cost for high-resolution images. To the best of our knowledge, the proposed focal attention provides the first mechanism to incorporate local and global attention in a single Transformer layer [2]. It can capture both short- and long-range interactions as standard self-attention but in a much more efficient and effective way, especially for high-resolution images.

## 3 Method

### 3.1 Model architecture

To accommodate high-resolution dense prediction tasks, we employ a multi-scale model architecture as in [60, 76, 43]. As shown in Fig. 2, an image $I \in \mathcal{R}^{H \times W \times 3}$ is first partitioned into patches of size $4 \times 4$, resulting in $\frac{H}{4} \times \frac{W}{4}$ visual tokens with dimension $4 \times 4 \times 3$. Then, we use a patch embedding layer, consisting of a convolutional layer with filter size and stride both equal to 4, to project these patches into hidden features with dimension $d$. We then pass this spatial feature map to the four stages of Focal Transformer blocks. In each stage $i \in \{1, 2, 3, 4\}$, the Focal Transformer block consists of $N_i$ Focal Transformer layers. After each stage, we use a patch embedding layer to reduce the spatial size of feature map by factor 2 and increase the feature dimension by 2. For image classification tasks, we take the average of the output from the last stage and send it to a classification layer. For object detection, the feature maps from the last 3 or all 4 stages are fed to a particular object detector head,

---

[2]A similar focal mechanism has been used in CNNs for NLP [27].

depending on the specific detection method we choose to use. The model capacity can be customized by varying the input feature dimension $d$ and the number of Focal Transformer layers.

Standard self-attention can capture both short- and long-range interactions at fine-grain, but suffers from high computational cost when it performs attention on high-resolution feature maps as noted in [76]. Take stage 1 in Fig. 2 as an example. For a feature map of size $\frac{H}{4} \times \frac{W}{4} \times d$, the complexity of self-attention is $\mathcal{O}((\frac{H}{4} \times \frac{W}{4})^2 d)$, resulting in an explosion of time and memory cost, considering that $\min(H, W)$ could be 800 or even larger for object detection. In the next section, we describe how we address this issue with the proposed focal attention mechanism.

## 3.2 Token-wise focal attention

Focal attention is proposed to make the Transformer layers suitable for encoding high-resolution input images. Instead of attending all tokens at fine-grain, we attend the fine-grain tokens only locally, but the summarized ones (i.e., the coarse-grained tokens generated by sub-window pooling, which is illustrated in Fig. 4 and will be described later) globally. As such, focal attention can cover the same amount of image regions as standard self-attention but with much less cost. In Fig. 3, we show the size of the receptive field for standard self-attention and our focal attention as a function of the number of attended tokens. For a given query position, by reducing the granularity of its surroundings based on their distance to the query, focal attention can have significantly larger receptive fields at the same cost measured by the number of visual tokens, compared to the standard self-attention mechanism.

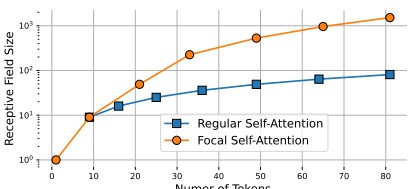

Figure 3: The size of receptive field (y-axis) as a function of the number of used visual tokens (x-axis) in regular (standard) self-attention and focal attention. When plotting the curve for focal attention, we increase the focal window size by 2 for each focal level up to the maximal window size of 8.

Theoretically, the focal attention mechanism enables global interaction with much less time and memory cost, because it attends a much smaller number of surrounding (summarized) tokens. In practice, however, extracting the surrounding tokens for each query position could incur high time cost since we need to duplicate the extraction of each token for all queries that the token surrounds. This issue had been extensively discussed in [56, 76, 43] and a common solution is to partition the input feature map into windows. Thus, in our Focal Transformers, we resort to performing focal attention at the window level. We elaborate the window-wise focal attention in the following.

### 3.2.1 Window-wise focal attention

Given a feature map of $x \in \mathcal{R}^{M \times N \times d}$ with spatial size $M \times N$, we first partition it into a grid of windows of size $s_p \times s_p$. Then, we extract the surroundings for each window rather than each individual token. The proposed window-wise focal attention is illustrated in Fig. 4. To clarify, we first define three terms:

- **Focal level** $L$ refers to the granularity level at which we extract the tokens for focal attention.
- **Focal window size** $s_w^l$ is the size of sub-window on which the summarized tokens are formed via sub-window pooling at granularity level of $l \in \{1, ..., L\}$.
- **Focal region size** $s_r^l$ denotes the number of sub-windows that are filled up horizontally (or vertically) in an attended region at level $l$.

Now, we detail how window-wise focal attention works in the following two steps, sub-window pooling and attention computing.

**Sub-window pooling**. Consider input feature map $x \in \mathcal{R}^{M \times N \times d}$, where $M \times N$ is the spatial dimension and $d$ the feature dimension. We perform sub-window pooling for all $L$ levels. At focal level $l$, we first split the input feature map $x$ into a grid of sub-windows with size $s_w^l \times s_w^l$. Then we use a linear projection layer $f_p^l$ to pool the sub-windows spatially by

$$x^l = f_p^l(\hat{x}) \in \mathcal{R}^{\frac{M}{s_w^l} \times \frac{N}{s_w^l} \times d}, \quad \hat{x} = \text{Reshape}(x) \in \mathcal{R}^{(\frac{M}{s_w^l} \times \frac{N}{s_w^l} \times d) \times (s_w^l \times s_w^l)}. \tag{1}$$

The pooled feature maps $\{x^l\}_1^L$ at different levels $l$ provide rich information at both fine-grain and coarse-grain. Since we set $s_w^l = 1$ for the first focal level which has the same granularity as the input

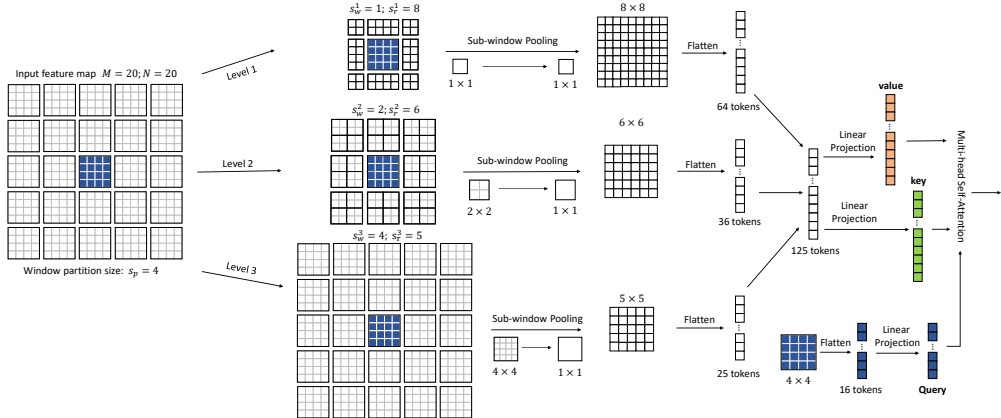

Figure 4: An illustration of focal attention at window level. Each of the square cells represents a visual token that is either from the original feature map or a summarized token formed by sub-window pooling. Suppose we have an input feature map of size $20 \times 20$. We first partition it into $5 \times 5$ windows of size $4 \times 4$. Take the $4 \times 4$ blue window in the middle as the query set, we extract its surrounding tokens at three granularity levels as its keys and values. For the first level, we extract the $8 \times 8$ tokens which are closest to the blue window at the finest grain. At the second level, we expand the attention region and pool the surrounding $2 \times 2$ sub-windows to form summarized tokens, which results in $6 \times 6$ summarized tokens. At the third level, we attend a larger region covering the whole feature map and pool $4 \times 4$ sub-windows, which leads to $5 \times 5$ summarized tokens. Finally, these three levels of tokens are concatenated to compute the keys and values for the $4 \times 4 = 16$ tokens (queries) in the blue window.

feature map, there is no need to perform any sub-window pooling. Considering that the focal window size is usually very small (7 maximally in our settings), the number of extra parameters introduced by sub-window pooling is negligible.

**Attention computing**. Once we obtain the pooled feature maps $\{x^l\}_1^L$ at all $L$ levels, we compute the query at the first level, and key and value for all levels using three linear projection layers $f_q$, $f_k$ and $f_v$, respectively, as

$$Q = f_q(x^1), \quad K = \{K^l\}_1^L = f_k(\{x^1, ..., x^L\}), \quad V = \{V^l\}_1^L = f_v(\{x^1, ..., x^L\}). \tag{2}$$

To perform focal attention, we need to first extract the surrounding tokens for each query token in the feature map. As mentioned earlier, tokens inside a window partition $s_p \times s_p$ share the same set of surroundings. For the queries inside the $i$-th window $Q_i \in \mathcal{R}^{s_p \times s_p \times d}$, we extract the $s_r^l \times s_r^l$ keys and values from $K^l$ and $V^l$ surrounding the window which the query lies in, and then gather the keys and values from all $L$ levels to obtain $K_i = \{K_i^1, ..., K_i^L\} \in \mathcal{R}^{s \times d}$ and $V_i = \{V_i^1, ..., V_i^L\} \in \mathcal{R}^{s \times d}$, where $s$ is the sum of focal regions from all levels, i.e., $s = \sum_{l=1}^L (s_r^l)^2$. Note that a *canonical* implementation of focal attention following Fig. 1 requires to exclude the overlapped regions across different levels. In our implementation, we intentionally keep them in order to capture the pyramid information for the overlapped regions. Finally, we follow [43] to include a relative position bias and compute the focal attention for $Q_i$ by

$$\text{Attention}(Q_i, K_i, V_i) = \text{Softmax}(\frac{Q_i K_i^T}{\sqrt{d}} + B)V_i, \tag{3}$$

where $B = \{B^l\}_1^L$ is the learnable relative position bias. It consists of $L$ subsets for $L$ focal levels. Similar to [43], for the first level, we parameterize it as $B^1 \in \mathcal{R}^{(2s_p-1) \times (2s_p-1)}$, considering that the horizontal and vertical position ranges are both in $[-s_p + 1, s_p - 1]$. For the other focal levels, considering that they have different granularity with respect to the queries, we treat all the queries inside a window equally and use $B^l \in \mathcal{R}^{s_r^l \times s_r^l}$ to represent the relative position bias between the query window and each of $s_r^l \times s_r^l$ summarized tokens. Since the focal attention for each window can be performed independent of the others, we can compute Eq. (3) in parallel. Once we obtain attention scores for the whole input feature map, we send them to LayerNorm and the MLP block.

| | Output Size | Layer Name | Focal-Tiny | Focal-Small | Focal-Base |
|---|---|---|---|---|---|
| stage 1 | $56 \times 56$ | Patch Embedding | $p_1 = 4; c_1 = 96$ | $p_1 = 4; c_1 = 96$ | $p_1 = 4; c_1 = 128$ |
| | $56 \times 56$ | Transformer Block | $\begin{bmatrix} s_{w,r}^0 = \{1,13\} \\ s_{w,r}^1 = \{7,7\} \end{bmatrix} \times 2$ | $\begin{bmatrix} s_{w,r}^0 = \{1,13\} \\ s_{w,r}^1 = \{7,7\} \end{bmatrix} \times 2$ | $\begin{bmatrix} s_{w,r}^0 = \{1,13\} \\ s_{w,r}^1 = \{7,7\} \end{bmatrix} \times 2$ |
| stage 2 | $28 \times 28$ | Patch Embedding | $p_2 = 2; c_2 = 192$ | $p_2 = 2; c_2 = 192$ | $p_2 = 2; c_2 = 256$ |
| | $28 \times 28$ | Transformer Block | $\begin{bmatrix} s_{w,r}^0 = \{1,13\} \\ s_{w,r}^1 = \{7,5\} \end{bmatrix} \times 2$ | $\begin{bmatrix} s_{w,r}^0 = \{1,13\} \\ s_{w,r}^1 = \{7,5\} \end{bmatrix} \times 2$ | $\begin{bmatrix} s_{w,r}^0 = \{1,13\} \\ s_{w,r}^1 = \{7,5\} \end{bmatrix} \times 2$ |
| stage 3 | $14 \times 14$ | Patch Embedding | $p_3 = 2; c_3 = 384$ | $p_3 = 2; c_3 = 384$ | $p_3 = 2; c_3 = 512$ |
| | $14 \times 14$ | Transformer Block | $\begin{bmatrix} s_{w,r}^0 = \{1,13\} \\ s_{w,r}^1 = \{7,3\} \end{bmatrix} \times 6$ | $\begin{bmatrix} s_{w,r}^0 = \{1,13\} \\ s_{w,r}^1 = \{7,3\} \end{bmatrix} \times 18$ | $\begin{bmatrix} s_{w,r}^0 = \{1,13\} \\ s_{w,r}^1 = \{7,3\} \end{bmatrix} \times 18$ |
| stage 4 | $7 \times 7$ | Patch Embedding | $p_4 = 2; c_4 = 768$ | $p_4 = 2; c_4 = 768$ | $p_4 = 2; c_4 = 1024$ |
| | $7 \times 7$ | Transformer Block | $\begin{bmatrix} s_{w,r}^0 = \{1,7\} \\ s_{w,r}^1 = \{7,1\} \end{bmatrix} \times 2$ | $\begin{bmatrix} s_{w,r}^0 = \{1,7\} \\ s_{w,r}^1 = \{7,1\} \end{bmatrix} \times 2$ | $\begin{bmatrix} s_{w,r}^0 = \{1,7\} \\ s_{w,r}^1 = \{7,1\} \end{bmatrix} \times 2$ |

Table 1: Model configurations for Focal Transformers. We use three configurations with different model capacities: Focal-Tiny, Focal-Small and Focal-Base.

### 3.2.2 Complexity analysis

We analyze the computational complexity for the two steps of focal attention described above. For the input feature map $x \in \mathcal{R}^{M \times N \times d}$, we have $\frac{M}{s_w^l} \times \frac{N}{s_w^l}$ sub-windows at focal level l. For each sub-window, the pooling operation in Eq.1 has the complexity of $\mathcal{O}((s_w^l)^2 d)$. Aggregating all sub-windows brings us $\mathcal{O}((MN)d)$. Then for all focal levels, we have the complexity of $\mathcal{O}(L(MN)d)$ in total, which is independent of the sub-window size at each focal level. Regarding the attention computation in Eq. 3, the computational cost for a query window $s_p \times s_p$ is $\mathcal{O}((s_p)^2 \sum_l (s_r^l)^2 d)$, and $\mathcal{O}(\sum_l (s_r^l)^2 (MN)d)$ for the whole input feature map. To sum up, the overall computational cost for focal attention is $\mathcal{O}((L + \sum_l (s_r^l)^2)(MN)d)$. In an extreme case, one can set $s_r^L = 2 \times \max(M, N)/s_w^L$ to ensure a global receptive field for all queries (including both corner and middle queries) in this layer.

### 3.3 Model configurations

For fair comparison, we consider three network configurations for Focal Transformers, following [60, 64, 43]. Specifically, we follow the design of the Tiny, Small and Base models in Swin Transformer [43], as shown in Table 1. Our models take $224 \times 224$ images as inputs and the window partition size is set to 7 to make our models comparable to Swin Transformers. For the focal attention layer, we introduce two levels, one for fine-grained local attention and the other for coarse-grained global attention. Except for the last stage, the focal region size is set to 13 for the window partition size of 7, which means that we expand 3 tokens for each window partition. For the last stage, since the whole feature map is $7 \times 7$, the focal region size at level 0 is set to 7, which is sufficient to cover the entire feature map. For the coarse-grained global attention, we set its focal window size the same as the window partition size 7, but gradually decrease the focal region size to get $\{7, 5, 3, 1\}$ for the four stages, respectively. For the patch embedding layer, the spatial reduction ratio $p_i$ for the four stages are all $\{4, 2, 2, 2\}$. Note that Focal-Base has a higher hidden dimension $c_i$, compared to Focal-Tiny and Focal-Small.

## 4 Experiments

### 4.1 Image classification on ImageNet-1K

We compare different methods on ImageNet-1K [19]. For fair comparison, we follow the training recipes in [55, 60]. All models are trained for 300 epochs with batch size 1024. The initial learning rate is set to $10^{-3}$ with 20 epochs of linear warm-up starting from $10^{-5}$. For optimization, we use AdamW [44] as the optimizer with a cosine learning rate scheduler. The weight decay is set to 0.05 and the maximal gradient norm is clipped to 5.0. We use the same set of data augmentation and regularization strategies used in [55] after excluding random erasing [82], repeated augmentation [4, 34] and exponential moving average (EMA) [48]. The stochastic depth drop rates are set to 0.2, 0.2 and 0.3 for our tiny, small and base models, respectively. During training, we crop images randomly to $224 \times 224$, while a center crop is used during evaluation on the validation set.

| Model | #Params. | FLOPs | Top-1 (%) |
|---|---|---|---|
| ResNet-50 [33] | 25.0 | 4.1 | 76.2 |
| DeiT-Small/16 [55] | 22.1 | 4.6 | 79.9 |
| PVT-Small [60] | 24.5 | 3.8 | 79.8 |
| ViL-Small [76] | 24.6 | 5.1 | 82.0 |
| CvT-13 [64] | 20.0 | 4.5 | 81.6 |
| Swin-Tiny [43] | 28.3 | 4.5 | 81.2 |
| Focal-Tiny (Ours) | 28.9 | 4.9 | **82.2** |
| ResNet-101 [33] | 45.0 | 7.9 | 77.4 |
| PVT-Medium [60] | 44.2 | 6.7 | 81.2 |
| CvT-21 [64] | 32.0 | 7.1 | 82.5 |
| ViL-Medium [76] | 39.7 | 9.1 | 83.3 |
| Swin-Small [43] | 49.6 | 8.7 | 83.1 |
| Focal-Small (Ours) | 51.1 | 9.4 | **83.6** |
| ResNet-152 [33] | 60.0 | 11.0 | 78.3 |
| ViT-Base/16 [21] | 86.6 | 17.6 | 77.9 |
| DeiT-Base/16 [55] | 86.6 | 17.5 | 81.8 |
| PVT-Large [60] | 61.4 | 9.8 | 81.7 |
| ViL-Base [76] | 55.7 | 13.4 | 83.2 |
| Swin-Base [43] | 87.8 | 15.4 | 83.4 |
| Focal-Base (Ours) | 89.8 | 16.4 | **84.0** |

Table 2: Comparison of image classification on ImageNet-1K for different models. Except for ViT-Base/16, all other models are trained and evaluated on $224 \times 224$ resolution.

| Backbone | RetinaNet | Mask R-CNN | |
|---|---|---|---|
| | $AP^b$ | $AP^b$ | $AP^m$ |
| ResNet-50 [33] | 36.3 | 38.0 | 34.4 |
| PVT-Small | 40.4 | 40.4 | 37.8 |
| ViL-Small [76] | 41.6 | 41.8 | 38.5 |
| Swin-Tiny [43] | 42.0 | 43.7 | 39.8 |
| Focal-Tiny (Ours) | **43.7** (+1.7) | **44.8** (+1.1) | **41.0** (+1.3) |
| ResNet-101 [33] | 38.5 | 40.4 | 36.4 |
| ResNeXt101-32x4d [67] | 39.9 | 41.9 | 37.5 |
| PVT-Medium [60] | 41.9 | 42.0 | 39.0 |
| ViL-Medium [76] | 42.9 | 43.4 | 39.7 |
| Swin-Small [43] | 45.0 | 46.5 | 42.1 |
| Focal-Small (Ours) | **45.6** (+0.6) | **47.4** (+0.9) | **42.8** (+0.7) |
| ResNeXt101-64x4d [67] | 41.0 | 42.8 | 38.4 |
| PVT-Large [60] | 42.6 | 42.9 | 39.5 |
| ViL-Base [76] | 44.3 | 45.1 | 41.0 |
| Swin-Base [43] | 45.0 | 46.9 | 42.3 |
| Focal-Base (Ours) | **46.3** (+1.3) | **47.8** (+0.9) | **43.2** (+0.9) |

Table 3: Comparisons with CNN and Transformer baselines and SoTA methods on COCO object detection. The box mAP ($AP^b$) and mask mAP ($AP^m$) are reported for RetinaNet and Mask R-CNN trained with $1\times$ schedule. More detailed comparisons with $3\times$ schedule are in Table 4.

In Table 2, we summarize the results for baseline models and the state-of-the-art models on image classification task. We can see that Focal Transformers consistently outperform other methods with similar model sizes (#Params.) and computational complexities (GFLOPs). Specifically, Focal-Tiny improves over the Transformer baseline DeiT-Small/16 by 2.3%. Meanwhile, using the same model configuration (2-2-6-2) and a few extra parameters and computations, Focal-Tiny improves over Swin-Tiny by 1.0 point. For small and base models, Focal-Small with 51.1M parameters can reach 83.6% which is better than all the counterpart small and base models using much less parameters. By increasing the model size, Focal-Base model achieves 84.0%, surpassing all the other models with comparable parameters and FLOPs.

To compare with the large-scale models, we further build Focal-Large Transformer by increasing the hidden dimension in Focal-Base from 128 to 196 while keeping all the other hyperparameters the same. We follow the common practice to pretrain our Focal-Large Transformer on ImageNet-22K and transfer it to detection and segmentation tasks [64, 43].

## 4.2 Object detection and instance segmentation

We benchmark our models on object detection with COCO 2017 [42]. The pretrained models are used as visual backbones and then plugged into two representative pipelines, RetinaNet [41] and Mask R-CNN [32]. All models are trained on the 118k training images and the results are reported on 5K validation set. We use the two standard training schedules, $1\times$ with 12 epochs and $3\times$ with 36 epochs. For the $1\times$ schedule, we resize image's shorter side to 800 while keeping its longer side no more than 1,333. For the $3\times$ schedule, we use the multi-scale training strategy by randomly resizing its shorter side to the range of $[480, 800]$. Considering this higher input resolution, we adaptively increase the focal sizes at four stages to $(15, 13, 9, 7)$, to ensures that the focal attention covers more than half of the image region at the first two stages, and the whole image at the last two stages. With the focal size increased, the relative position biases are accordingly up-sampled to the corresponding sizes using bilinear interpolation. During training, we use AdamW [44] for optimization with initial learning rate $10^{-4}$ and weight decay 0.05. Similarly, we use 0.2, 0.3 and 0.5 stochastic depth drop rates to regularize the training for our Tiny, Small and Base models, respectively. Since Swin Transformer does not report the results on RetinaNet, we obtain the results by ourselves using their official code with the same hyper-parameters as that of Focal Transformers.

In Table 3, we show the performance for both CNN-based models and the current Transformer-based state-of-the-art models. The bbox mAP ($AP^b$) and mask mAP ($AP^m$) are reported. We see that Focal Transformers outperform the CNN-based models consistently with the gap of 4.8-7.1 points. Compared with the other methods which also use multi-scale Transformer architectures,

| Backbone | #Params | FLOPs | RetinaNet 3x schedule + MS | | | | | | Mask R-CNN 3x schedule + MS | | | | | |
|---|---|---|---|---|---|---|---|---|---|---|---|---|---|---|
| | (M) | (G) | $AP^b$ | $AP^b_{50}$ | $AP^b_{75}$ | $AP_S$ | $AP_M$ | $AP_L$ | $AP^b$ | $AP^b_{50}$ | $AP^b_{75}$ | $AP^m$ | $AP^m_{50}$ | $AP^m_{75}$ |
| ResNet50 [33] | 37.7/44.2 | 239/260 | 39.0 | 58.4 | 41.8 | 22.4 | 42.8 | 51.6 | 41.0 | 61.7 | 44.9 | 37.1 | 58.4 | 40.1 |
| PVT-Small[60] | 34.2/44.1 | 226/245 | 42.2 | 62.7 | 45.0 | 26.2 | 45.2 | 57.2 | 43.0 | 65.3 | 46.9 | 39.9 | 62.5 | 42.8 |
| ViL-Small [76] | 35.7/45.0 | 252/174 | 42.9 | 63.8 | 45.6 | 27.8 | 46.4 | 56.3 | 43.4 | 64.9 | 47.0 | 39.6 | 62.1 | 42.4 |
| Swin-Tiny [43] | 38.5/47.8 | 245/264 | 45.0 | 65.9 | 48.4 | 29.7 | 48.9 | 58.1 | 46.0 | 68.1 | 50.3 | 41.6 | 65.1 | 44.9 |
| Focal-Tiny (Ours) | 39.4/48.8 | 265/291 | **45.5** | **66.3** | **48.8** | **31.2** | **49.2** | **58.7** | **47.2** | **69.4** | **51.9** | **42.7** | **66.5** | **45.9** |
| ResNet101 [33] | 56.7/63.2 | 315/336 | 40.9 | 60.1 | 44.0 | 23.7 | 45.0 | 53.8 | 42.8 | 63.2 | 47.1 | 38.5 | 60.1 | 41.3 |
| ResNeXt101-32x4d [67] | 56.4/62.8 | 319/340 | 41.4 | 61.0 | 44.3 | 23.9 | 45.5 | 53.7 | 44.0 | 64.4 | 48.0 | 39.2 | 61.4 | 41.9 |
| PVT-Medium [60] | 53.9/63.9 | 283/302 | 43.2 | 63.8 | 46.1 | 27.3 | 46.3 | 58.9 | 44.2 | 66.0 | 48.2 | 40.5 | 63.1 | 43.5 |
| ViL-Medium [76] | 50.8/60.1 | 339/261 | 43.7 | 64.6 | 46.4 | 27.9 | 47.1 | 56.9 | 44.6 | 66.3 | 48.5 | 40.7 | 63.8 | 43.7 |
| Swin-Small [43] | 59.8/69.1 | 335/354 | 46.4 | 67.0 | 50.1 | 31.0 | 50.1 | 60.3 | 48.5 | 70.2 | 53.5 | 43.3 | 67.3 | 46.6 |
| Focal-Small (Ours) | 61.7/71.2 | 367/401 | **47.3** | **67.8** | **51.0** | **31.6** | **50.9** | **61.1** | **48.8** | **70.5** | **53.6** | **43.8** | **67.7** | **47.2** |
| ResNeXt101-64x4d [67] | 95.5/102 | 473/493 | 41.8 | 61.5 | 44.4 | 25.2 | 45.4 | 54.6 | 44.4 | 64.9 | 48.8 | 39.7 | 61.9 | 42.6 |
| PVT-Large[60] | 71.1/81.0 | 345/364 | 43.4 | 63.6 | 46.1 | 26.1 | 46.0 | 59.5 | 44.5 | 66.0 | 48.3 | 40.7 | 63.4 | 43.7 |
| ViL-Base [76] | 66.7/76.1 | 443/365 | 44.7 | 65.5 | 47.6 | 29.9 | 48.0 | 58.1 | 45.7 | 67.2 | 49.9 | 41.3 | 64.4 | 44.5 |
| Swin-Base [43] | 98.4/107 | 477/496 | 45.8 | 66.4 | 49.1 | 29.9 | 49.4 | 60.3 | 48.5 | 69.8 | 53.2 | 43.4 | 66.8 | 46.9 |
| Focal-Base (Ours) | 100.8/110.0 | 514/533 | **46.9** | **67.8** | **50.3** | **31.9** | **50.3** | **61.5** | **49.0** | **70.1** | **53.6** | **43.7** | **67.6** | **47.0** |

Table 4: COCO object detection and segmentation results with RetinaNet [41] and Mask R-CNN [33]. All models are trained with $3\times$ schedule and multi-scale inputs (MS). The numbers before and after "/" at column 2 and 3 are the model size and complexity for RetinaNet and Mask R-CNN, respectively.

Focal Transformers show substantial gains across all settings and metrics. Particularly, Focal Transformers brings 0.7-1.7 points of mAP against the current best approach Swin Transformer [43] at comparable settings. Different from the other multi-scale Transformer models, Focal Transformers can simultaneously enable short-range fine-grain and long-range coarse-grain interactions for each visual token, and thus capture richer visual contexts at each layer for better dense predictions. To have more comprehensive comparisons, we train all models using the $3\times$ schedule and show the detailed numbers for RetinaNet and Mask R-CNN in Table 4. As we can see, even with the $3\times$ schedule, Focal Transformers can still achieve 0.3-1.1 gain over Swin Transformer models in comparable settings.

**Comparison with large SoTA detection models.** We follow Swin Transformers to use HTC [11] as the detection method in that it reported SoTA performance on COCO detection when using Swin Transformer as the backbone. For fair comparison, we also use soft-NMS [5], instaboost [24] and a multi-scale training strategy with the shorter side in range $[400, 1400]$ and the longer side no more than 1600. We train the model using AdamW [44] with base learning rate 1e-4 and weight decay 0.1. The model is trained using the standard $3\times$ schedule. The box and mask mAPs on COCO validation set and test-dev are reported in Table 5, where both single-scale evaluation and multi-scale evaluation results are presented. Our Focal-Large model with multi-scale test achieves 58.1 box mAP and 50.9 mask mAP on mini-val set, which is better than the reported numbers for Swin-Large in [43]. When evaluating our model on the test-dev set, it achieves 58.4 box mAP and 51.3 mask mAP, which is slightly better than Swin Transformer. Note that because our model does not include the global self-attention layer used in Swin Transformer at the last stage, it has a smaller model size and fewer FLOPs. More recently, DyHead [17] achieves new SoTA on COCO, when combined with Swin-Large. We replace the Swin-Large model with the Focal-Large model, and use the same $2\times$ training schedule as in [17]. We report the box mAPs for both mini-val and test-dev. Focal-Large achieves **58.7** and **59.0** on mini-val and test-dev, respectively.

### 4.3 Semantic Segmentation

In addition to the instance segmentation results, we also evaluate our models on the semantic segmentation task which usually takes high-resolution input images and requires capturing long-range interactions. We benchmark our methods on ADE20K [83]. We use UperNet [65] as the segmentation method and Focal Transformers as the backbones. We train three models as Focal-Tiny, Focal-Small, Focal-Base, respectively. For all the models, we use a standard recipe that sets the input size to $512 \times 512$ and trains the model for 160k iterations with batch size 16. Table 6 shows the comparison results. We see that Focal-Tiny, Focal-Small and Focal-Base models consistently outperform Swin Transformers of the similar size in single-scale and multi-scale mIoUs.

**Comparison with large SoTA semantic segmentation models.** We use the pretrained Focal-Large model as the backbone for semantic segmentation. Follow the setting in [43], we use input image size $640 \times 640$ and train the model for 160k iterations with a batch size of 16. We set the initial learning to 6e-5 and use a polynomial learning rate decay. The weight decay is set to $0.01$. For

| Method | #Param | FLOPs | mini-val | | test-dev | |
|---|---|---|---|---|---|---|
| | | | $AP^b$ | $AP^m$ | $AP^b$ | $AP^m$ |
| X101-64x4d [67] | 155M | 1033G | 52.3 | 46.0 | - | - |
| EfficientNet-D7 [54] | 77M | 410G | 54.4 | - | 55.1 | - |
| GCNet* [7] | - | 1041G | 51.8 | 44.7 | 52.3 | 45.4 |
| ResNeSt-200 [75] | - | - | 52.5 | - | 53.3 | 47.1 |
| Copy-paste [28] | 185M | 1440G | 55.9 | 47.2 | 56.0 | 47.4 |
| BoTNet-200 [51] | - | - | 49.7 | - | - | - |
| SpineNet-190 [22] | 164M | 1885G | 52.6 | - | 52.8 | - |
| CenterNet2 [84] | - | - | - | - | 56.4 | - |
| Swin-L (HTC++) [43] | 284M | 1470G | 57.1 | 49.5 | 57.7 | 50.2 |
| Swin-L (DyHead) [17] | 213M | 965G | 56.2 | - | - | - |
| Swin-L† (HTC++) [43] | 284M | - | 58.0 | 50.4 | 58.7 | 51.1 |
| Swin-L† (DyHead) [17] | 213M | - | 58.4 | - | 58.7 | - |
| Swin-L† (QueryInst) [25] | - | - | 56.1 | - | 56.1 | - |
| Focal-L (HTC++) (Ours) | 265M | 1165G | 57.0 | 49.9 | - | - |
| Focal-L (DyHead) (Ours) | 229M | 1081G | 56.4 | - | - | - |
| Focal-L† (HTC++) (Ours) | 265M | - | 58.1 | **50.9** | 58.4 | **51.3** |
| Focal-L† (DyHead) (Ours) | 229M | - | **58.7** | - | **59.0** | - |

Table 5: Comparison with state-of-the-art methods on COCO object detection and instance segmentation. The numbers are reported on 5K val set and test-dev. Augmented HTC [11] (denoted by HTC++) and DyHead [17] are used as the detection methods. † means multi-scale evaluation.

| Backbone | Method | #Param | FLOPs | mIoU | +MS |
|---|---|---|---|---|---|
| ResNet-101 | DANet [45] | 69M | 1119G | 45.3 | - |
| ResNet-101 | ACNet [26] | - | - | 45.9 | - |
| ResNet-101 | DNL [69] | 69M | 1249G | 46.0 | - |
| ResNet-101 | UperNet [65] | 86M | 1029G | 44.9 | - |
| HRNet-w48 [53] | OCRNet [73] | 71M | 664G | 45.7 | - |
| ResNeSt-200 [75] | DLab.v3+ [12] | 88M | 1381G | 48.4 | - |
| Swin-T [43] | UperNet [65] | 60M | 945G | 44.5 | 45.8 |
| Swin-S [43] | UperNet [65] | 81M | 1038G | 47.6 | 49.5 |
| Swin-B [43] | UperNet [65] | 121M | 1188G | 48.1 | 49.7 |
| Twins-SVT-L [15] | UperNet [65] | 133M | - | 48.8 | 50.2 |
| MiT-B5 [66] | SegFormer [66] | 85M | - | 51.0 | 51.8 |
| ViT-L/16† [21] | SETR [80] | 308M | - | 50.3 | - |
| Swin-L‡ [43] | UperNet [65] | 234M | 3230G | 52.1 | 53.5 |
| ViT-L/16‡ [21] | Segmenter [52] | 334M | - | 51.8 | 53.6 |
| Swin-L‡ [43] | K-Net [77] | - | - | - | 54.3 |
| Swin-L‡ [43] | PatchDiverse [29] | 234M | - | 53.1 | 54.4 |
| VOLO-D5 [72] | UperNet [65] | - | - | - | 54.3 |
| Focal-T (Ours) | UperNet [65] | 62M | 998G | 45.8 | 47.0 |
| Focal-S (Ours) | UperNet [65] | 85M | 1130G | 48.0 | 50.0 |
| Focal-B (Ours) | UperNet [65] | 126M | 1354G | 49.0 | 50.5 |
| Focal-L‡ (Ours) | UperNet [65] | 240M | 3376G | **54.0** | **55.4** |

Table 6: Comparison with SoTA methods for semantic segmentation on ADE20K [83] val set. Single- and multi-scale evaluations are reported in the last two columns. ‡ means ImageNet-22K is used as the pretraining dataset.

| Model | W-Size | FLOPs | Top-1 (%) | $AP^b$ | $AP^m$ |
|---|---|---|---|---|---|
| Swin-Tiny | 7 | 4.5 | 81.2 | 43.7 | 39.8 |
| | 14 | 4.9 | 82.1 | 44.0 | 40.5 |
| Focal-Tiny | 7 | 4.9 | 82.2 | 44.9 | 41.1 |
| | 14 | 5.2 | 82.3 | 45.5 | 41.5 |

Table 7: Impact of different window sizes (W-Size). We alter the default size 7 to 14 and observe consistent improvements for both methods.

| Model | W-Shift | Top-1 (%) | $AP^b$ | $AP^m$ |
|---|---|---|---|---|
| Swin-Tiny | - | 80.2 | 38.8 | 36.4 |
| | ✓ | 81.2 | 43.7 | 39.8 |
| Focal-Tiny | - | 82.2 | 44.8 | 41.0 |
| | ✓ | 81.9 | 44.9 | 41.1 |

Table 8: Impact of window shift (W-Shift) on Swin Transformer and Focal Transformer. Tiny models are used.

multi-scale evaluation, we use the same scaling ratios $[0.5, 0.75, 1.0, 1.25, 1.5, 1.75]$ as in previous works. The results in Table 6 show that Focal-Large achieves significantly better performance than Swin-Large. In both single-scale and multi-scale evaluations, Focal-Large leads to more than 1 point mIoU improvement, creating new SoTA for semantic segmentation on ADE20K.

### 4.4 Ablation studies

We conduct a series of ablation studies to inspect the model's capacity from different aspects. We use Focal-Tiny and the image classification and object detection tasks.

**Effect of varying the window size.** We have demonstrated that it is crucial to model both short- and long-range interactions. Thus, a related question is whether increasing the window size helps as it leads to a larger receptive field. Table 7 shows the performance of Swin-Tiny and Focal-Tiny with window sizes 7 and 14. Clearly, a larger window size is beneficial for both methods measured in all three metrics, and Focal-Tiny consistently outperforms Swin-Tiny in both window sizes. Comparing the second and third row, we find that Focal-Tiny outperforms Swin-Tiny even with a smaller window size (7 *v.s.* 14). We suspect that the gain is attributed to our focal attention's superior capability of capturing long-range dependencies among visual tokens.

**The necessity of window shift.** In Swin Transformer [43], window shift is proposed to capture cross-window interactions between two successive layers. In contrast, visual tokens in Focal Transformers can always communicate with each other across windows at both fine- and coarse-grain. Thus, it is interesting to investigate whether adding window shift to Focal Transformers can lead to any improvement. To answer the question, we remove window shift from Swin Transformer while adding it to Focal Transformers. As shown in Table 8, Swin Transformer shows a severe degradation after removing the window shift. However, adding window shift to Focal Transformer hurts classification performance. The result indicates that window shift is unnecessary for Focal Transformers. While in

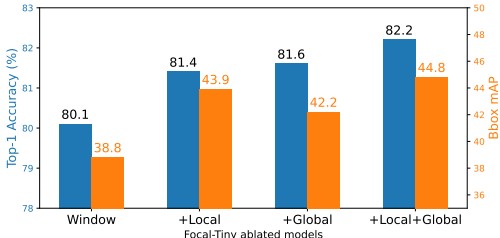

Figure 5: Ablating Focal-Tiny model by adding local, global and both interactions, respectively. Blue bars are image classification results and orange bars object detection results. This figure is better viewed in color.

| Depths | Model | #Params. | FLOPs | Top-1 (%) | $AP^b$ | $AP^m$ |
|---|---|---|---|---|---|---|
| 2-2-2-2 | Swin | 21.2 | 3.1 | 78.7 | 38.2 | 35.7 |
| | Focal | 21.7 | 3.4 | 79.9 | 40.5 | 37.6 |
| 2-2-4-2 | Swin | 24.7 | 3.8 | 80.2 | 41.2 | 38.1 |
| | Focal | 25.4 | 4.1 | 81.4 | 43.3 | 39.8 |
| 2-2-6-2 | Swin | 28.3 | 4.5 | 81.2 | 43.7 | 39.8 |
| | Focal | 29.1 | 4.9 | 82.2 | 44.8 | 41.0 |

Table 9: Impact of the change of model depth. We gradually reduce the number of transformer layers at the third stage from original 6 to 4 and further 2. Our Focal Transformers has much slower drop rate than Swin Transformer.

Swin Transformers, there should always be an even number of layers in each stage for the alternative window shift operation, Focal Transformers do not have such a constraint.

**Contributions of local and global interactions.** To investigate the relative contributions of capturing local fine-grain and global coarse-grain interactions in Focal Transformers, we have developed several variants of Focal-Tiny: a) Focal-Tiny-Window merely performs attention inside each window; b) Focal-Tiny-Local attends the additional fine-grain surrounding tokens and c) Focal-Tiny-Global attends the extra coarse-grain summarized tokens. We train these models using the same setting as Focal-Tiny and report their performance on image classification and object detection using Mask R-CNN $1\times$ schedule. As shown in Fig. 5, Focal-Tiny-Window suffers from a significant performance drop on both image classification (82.2→80.1) and object detection (44.8→38.3). This is expected since the communication across windows is completely cut off at each Transformer layer. After we enable either the local fine-grain or global coarse-grain interactions (middle two columns), we observe significant performance boost. When we combine short- and long-range interactions, we observe additional improvements on both tasks. This implies that these two type of interactions are complementary and both are beneficial to model performance.

**Model capacity against model depth.** Focal attention allows a Transformer model to capture short- and long-range interactions at each Transformer layer. An interesting question is whether Focal Transformers need fewer layers to obtain a similar modeling capacity as the Transformer models that does not use focal attention, such as Swin Transformer. To answer this question, we conduct an experiment by training a series of Swin-Tiny and Focal-Tiny models by varying the number of Transformer layers at stage 3. As shown in Table 9, Focal-Tiny outperforms Swin-Tiny consistently with the same depth. More importantly, using fewer layers, Focal-Tiny can sometimes achieve comparable or even better performance than Swin Transformer. For example, Focal-Tiny with (2-2-4-2) achieves 81.4 on image classification which is better than Swin-Tiny with (2-2-6-2).

## 5 Conclusion

In this paper, we have presented a new focal attention mechanism that enables efficient long-range interactions in Vision Transformers. Different from previous works, it performs the local attention at fine-grain and global attention at coarse-grain, providing an effective way of capturing both short- and long-range context with a manageable computational cost. By applying focal attention into a multi-scale Transformer architecture, we propose Focal Transformers as general-purpose backbones for a wide range of dense vision tasks. A comprehensive empirical study shows that our Focal Transformers outperform the SoTA Vision Transformers on a range of vision tasks including image classification, object detection and segmentation.

**Limitations and future work**. Although our experiments show that focal attention can significantly boost the performance on image classification and dense prediction tasks, focal attention does introduce extra computational and memory cost, since each query token needs to attend more (summarized) tokens in addition to tokens inside a window. A cost-effective implementation of Focal Transformer is necessary to make it more applicable to many real-world scenarios. This study focuses on incorporating focal attention into multi-scale Vision Transformers for CV tasks. However, we notice that focal attention is an effective sparse attention mechanism that is widely applicable to all attention-based neural network models that are developed for processing natural language, images, videos etc. This is an exciting future direction.

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
