# OpenReview forum: "Focal Attention for Long-Range Interactions in Vision Transformers"
_NeurIPS.cc/2021/Conference — NeurIPS 2021 Spotlight_

### Official Review · Reviewer_NEKg · 2021-07-15

**Rating:** 6
**Confidence:** 5

**Summary:**

This paper proposed a new attention mechanism called focal attention to enable efficient long-range interactions in vision transformer. By plugging the proposed module into a multi-scale vision transformer model,  a new focal transformer was designed and its superiority was demonstrated over the state-of-the-art approaches on both image classification and object detection tasks.

**Ethics Review Area:**

["I don’t know"]

**Limitations And Societal Impact:**

1. There are limited discussions on model efficiency. I suggest the authors to add more experimental comparisons with Sparse Transformer or some other efficient transformer methods.
2. The focal windows and region partition operations do not bring any additional flops but may be time-consuming.  It should better add throughoutput or latency results in Table2.
3. Figure 4 shows three fine to coarse attention levels with short and long range interaction.  However, in the actual model configuration the attention only has two level and mismatch Figure4. Why dose this happen?  The authors should do ablation study about the number of level including acc and latency.
4. In table 4 stage1-2, s^1_{w} * s^1_{r} is larger than input resolution, e.g, 7x5>28. How do you handle this situation.

**Main Review:**

1. This paper is well written, and has no obvious mistakes.
2. For a vision transformer model, how to reduce the computational overhead while keeping ability of capturing the long-range dependencies is still an important and interesting research topic.
3. The experiments are conducted on various benchmarks on image classification and object detection, with extensive ablation studies.


**Time Spent Reviewing:**

2h

---

> ### Author Response · Authors · 2021-08-10
> **Thanks for your valuable comments**
>
> First of all, we would like to thank the reviewer for pointing out the strengths of our work. We also thank the reviewer for the valuable comments and suggestions! We address the concerns one by one below.
>
> **Q 1&2. Model efficiency and throughputs.**
>
> A 1&2: Thanks for pointing this out. Below we report the throughput comparisons for different models below. All throughputs are measured on 1 V100 GPU with batch size 128 and input image resolution 224x224.
>
> | Model       | Top-1 Acc | GFLOPs | Throughput (imgs/s)  |
> | --- | --- | --- | --- |
> DeiT-Small/16 |      79.8       |  4.6     | 939
> PVT-Small        |      79.8       | 3.8      | 794
> CvT-13              |      81.6       | 4.5      | 746
> ViL-Small          |      82.0       | 5.1      | 397
> Swin-Tiny         |     81.2       | 4.5       | 760
> Focal-Tiny        |     82.2       | 4.9       | 319
> ||
> PVT-Medium  | 81.2        | 6.7      | 517
> CvT-21             | 82.5        | 7.1      | 480
> ViL-Medium    | 83.3        | 9.1      | 251
> Swin-Small      | 83.1        | 8.7      | 435
> Focal-Small     | 83.5        | 9.1      | 192
> ||
> ViT-Base/16    | 77.9        | 17.6     | 291
> Deit-Base/16  | 81.8        | 17.6     | 291
> PVT-Large       | 81.7        | 9.8       | 352
> ViL-Base          | 83.2        | 13.4     | 145
> Swin-Base       | 83.4        | 15.4     | 291
> Focal-Base      | 83.8        | 16.0     | 138
>
> In the above table, we do notice that Focal Transformers are slower than their counterparts except for ViL. We refer the reviewer to our response to Reviewer HEkP above for more comparisons on higher resolution inputs. Based on our investigations on different ablated models, we found the throughput decrease is because of two reasons: 1) the overhead to extract the fine-grain and coarse-grain tokens at each transformer layer; 2) the extra computation after adding more keys and values. By measuring the speed for each of the ablated Focal-Tiny models in Figure 5 of our main submission, we obtain the numbers in the below table. We further factorize the time cost of key and value extraction and attention computation by implementing pseudo versions of Focal-Tiny-local and Focal-TIny-global which retain the same number of keys and values by appending pseudo tokens instead of extracting them from the (pooled) feature map. As we can see, local fine-grained local attention leads to more computational cost than coarse-grained global attention. By comparing them with the corresponding pseudo versions, we can see that extracting fine-grained local tokens and coarse-grained global tokens both introduce more overheads than the attention computation. Moreover, extracting fine-grained tokens is relatively heavier than extract coarse-grained tokens.
>
> | Model                                    | Top-1 Acc | GFLOPs   | Throughput (imgs/s) |
> | --- | --- | --- | --- |
> Focal-Tiny                             | 82.2           | 4.92         | 319
> Focal-Tiny-local                    | 81.4           | 4.90         | 417
> Focal-Tiny-local-pseudo     | n/a             | 4.90         | 636
> Focal-Tiny-global                 | 81.6            | 4.59        | 516
> Focal-Tiny-global-pseudo   | n/a             | 4.59         | 633
> Focal-Tiny-window              | 80.1            | 4.49        | 731
>
> Based on the above comparisons, we notice the fine-grained local attention brings most of the extra time cost, and it is mainly due to the sub-optimal implementation of extracting surrounding fine-grained key and value tokens for each window. Currently, we implement this using rolling operations, which is two times faster than the conventional unfolding operation. However, we find this customized implementation is still under-optimized compared with a cuda kernel as used in the convolution layer, especially for the earlier stage which has relatively higher feature resolution. We believe when an efficient cuda kernel implementation is available, this extra time cost can be reduced significantly to approach the gap of FLOPs between different methods. Before that, we empirically studied how our models perform when removing a few time-consuming fine-grained local attentions. Specifically, we tried two variants, one is removing the fine-grained local attention at the first stage, the other one is removing local attention at all odd layers.  As we can see from the table below, these two variants have higher throughputs and slightly lower Top-1 accuracy, but still higher than the Swin-Transformer counterparts. These results imply that we may not need to have fine-grain local attention at all layers and all stages. We will have more systematic studies on this aspect.
>
> | Model                                                                      | Top-1 Acc | GFLOPs | Throughput (imgs/s)  |
> | --- | --- | --- | --- |
> Focal-Tiny                                                               |     82.2       | 4.92        |  319
> Focal-Tiny (no local fine-grain at first stage)    |     82.1       | 4.77        | 388
> Focal-Tiny (local fine-grain at even layers)       |     81.9      | 4.75         | 399
> ||
> Focal-Small                                                               |    83.5       |  9.12       | 192
> Focal-Small (no local fine-grain at first stage)    |   83.5       |  8.98      | 217
> Focal-Small (local fine-grain at even layers)       |   83.3       |  8.85      | 240
> ||
> Focal-Base                                                          |    83.8       | 16.04    | 138
> Focal-Base (no local fine-grain at first stage)     |    83.7       | 15.80   | 154
> Focal-Base (local fine-grain at even layers)        |    83.5       |  15.56  | 172
>
> **Q3. Comparing three focal levels with two focal levels used in the paper.**
>
> A3: Thanks for raising this question! The reason why we chose to use two focal levels in all our experiments is that we observed introducing the extra focal level 2 can slightly improve the performance in some cases but brings extra computational overheads in the meantime. Below we assume the finest level as focal level 1 and the coarsest level as focal level 3 and study whether adding the middle focal level can help improve the performance.
>
> | Model |  | focal level 1 | focal level 2 | focal level 3 | Top-1 acc | \#Params. | GFLOPs | Throughputs |
> | --- | --- | --- | --- | --- | --- | --- | --- | --- |
> Focal-Tiny | | - | - | yes      | 81.6   | 28.7 | 4.59  | 516
> Focal-Tiny | | - | yes | yes | 81.8   | 28.7 | 4.61   | 388
> Focal-Tiny | | yes | - | yes |  82.2 | 29.1 | 4.91 |  319
> Focal-Tiny | | yes | yes | yes | 82.2 | 29.2 | 4.93 | 269
>
> As we can see from the table, when focal level 1 is missed, adding focal level 2 can improve the performance by 0.2% with the Focal-Tiny model. However, when we use focal level 1, adding focal level 2 cannot further have sensible improvement. We suspect there may be two reasons. First, as we mentioned in lines 159-161, we did not exclude the overlapped tokens across different focal levels. This means that the pooled window tokens at focal level 2 actually have a lot of overlaps to focal levels 1 and 3. This redundancy may explain why adding focal level 2 can improve the performance when merely using focal 3 but not for the combination of focal levels 1 and 3. Second, as we mentioned in our submission, we used the same window size=7 for a fair comparison with Swin Transformers. However, this odd window size hinders us to get a good alignment between each 7x7 window and the sub-windows at focal level 2 (because we need to halve the window size for pooling). We need to perform either padding or trimming for the original feature map so that the number of pooled windows at focal level 2 exactly matches the required one for each window-wise focal attention. We think this misalignment may introduce some noises to the focal self-attention which dismisses the potential benefits. For comprehensiveness, we will add this ablation study in our revision. Moreover, to verify our assumption, we will study whether focal level 2 can bring us some improvements when the windows are well-aligned for focal self-attention, e.g. we change the window size from 7 to 8. We will post the results during the discussion when the experiments are completed.
>
> **Q4: Confusion about handling size mismatch in focal attention computation.**
>
> A4: Thanks for raising this confusion! Given the feature map at a certain stage, e.g., 28x28 at the second stage, we first perform window pooling with s^l_{w}=7 to get 4x4 feature map as described at lines 142-149, then we use unfold operation with (filter_size=5, stride=1, padding=2) to perform the unfolding to get 5x5=25 window tokens for each of these 4x4=16 windows. As such, each of 7x7=49 tokens in a window shares the surrounding 25 window tokens when performing self-attention. Note that we always mask out the padding tokens during unfolding for precise self-attention computations. This strategy is used across all layers and all stages in our Focal Transformers, and the padding size is set to s^l_{r} //2 to adapt to different focal region sizes at different stages.

---

> ### Author Response · Authors · 2021-08-20
> **One more experimental result on number of focal levels**
>
> We thank the reviewer again for the great suggestion of comparing two focal levels and three focal levels.
>
> In our previous response, we had reported the comparison between two and three focal levels with window size=7. We did not observe noticeable improvement when adding the second focal level into our current Focal-Tiny model. We suspect one of the two reasons is that the odd window size 7 makes it hard to align the pooled window tokens at the second level to a window. Hence, we change it to an even number 8. Note that changing window size to 8 can ensure the correct alignment but requires padding the feature map at each layer so that it can be divided by 8. Below we report the performance for our Focal-Tiny models using two and three focal levels with window size=8.
>
> | Model | window size | focal levels | Top-1 Acc. | #params. | GFLOPs | Throughput (imgs/s) |
> | --- | --- |  --- | --- |  --- | --- | --- |
> | Focal-Tiny | 8 | 2 | 82.16 | 29.20 | 4.99 | 282 |
> | Focal-Tiny | 8 | 3 | 82.26 | 29.22 | 5.03 | 226 |
> ||
>
> According to the above table, adding the second focal level does bring minor improvement to the original Focal-Tiny model. But still, the improvement is minor with respect to the extra overhead in terms of throughput. Along with our previous results, we can conclude that adding a second focal level can only bring minor improvement because the region is already covered by focal level 1 and 3 regarding the receptive field. Hence, we believe the gain brought by our proposed focal self-attention is mainly attributed to its ability to model both short- and long-term interactions across different visual tokens, rather than the pyramid of the feature map. As a result, we recommend using two focal levels as in our current Focal Transformers.

---

### Official Review · Reviewer_HEkP · 2021-07-16

**Rating:** 6
**Confidence:** 5

**Summary:**

This paper proposes Focal Transformer, which captures both local and global attention based on their design. It also follows the pyramid architecture so their model can be easily extended to detection and segmentation tasks. The result on image classification detection and segmentation are very promising.

**Limitations And Societal Impact:**

Yes

**Main Review:**

Overall, the main contribution of this paper is proposing a new attention module for multi-level trasnformer, which seems efficient and can capture both local and global information. Then the architecture heavily follows PVT and Swin Transformer.

About performance, it is slightly better than Swin Transformer, which is reasonable because swin do not have global attention.

Advantages:

1. clear paper writing, which is easy to follow.

2. strong performance on several tasks.

3. reasonable story to design an efficient self-attention but consider both global and local region.

Weakness:

1. I have some concerns about the inference speed of this method. If I have a misunderstanding please correct me. Mainly because calculating focal attention seems not very straightforward. I understand the theoretical computation complexity is efficient but it is not clear of real running speed. **It is necessary to report the speed and compare with other methods** such as Swin and PVT, especially on object detection and segmentation which need a high-resolution image as input.

2. In Table2 when scaling model size from small to base, the accuracy stops to increase. So I wonder does this model's performance upper bound is lower than other methods? Because ViT can consistently improve the performance when enlarging the model size even the model size is super large.


**Time Spent Reviewing:**

2

---

> ### Author Response · Authors · 2021-08-10
> **Thanks for your valuable comments**
>
> First of all, we would like to thank the reviewer for pointing out the contributions of our work. We also thank the reviewer for the valuable comments! We address the concerns one by one below.
>
> **Q1. Model efficiency comparison.**
>
> A1: Thanks for raising this concern! Below we report the throughputs for our Focal Transformers with different model sizes. For comparison, we also report the throughputs for PVT, Swin Transformer, Vision Longformer, CvT, etc. To compare the image classification efficiency, all throughputs are measured on 1 V100 GPU with batch size 128 and input image resolution 224x224. We also add the Top-1 accuracy and corresponding GFLOPs. Moreover, we measure the throughputs for higher input image resolutions 448x448 and 896x896 to simulate the object detection scenario. Note that we intend to **not** measure the full object detection models because all models share the same FPN/RPN/ROI head, which will discount the differences among different models.
>
> | Model               | Top-1 Acc | GFLOPs |    224x224  | 448x448 | 896x896  |
> |--------|--------|--------|--------| -------| -------|
> DeiT-Small/16 |      79.8       |  4.6     | 939            |  101          | 20
> PVT-Small        |      79.8       | 3.8      | 794            |  172          | 31
> CvT-13              |      81.6       | 4.5      | 746            |  125         | 14
> ViL-Small          |      82.0       | 5.1      | 397            |  87           | 17
> Swin-Tiny         |     81.2       | 4.5       | 760            |  189         | 48
> Focal-Tiny        |     82.2       | 4.9       | 319            |   105        | 27
> ||
> PVT-Medium  | 81.2        | 6.7      | 517                | 111          | 20
> CvT-21             | 82.5        | 7.1      | 480                | 85            | 10
> ViL-Medium    | 83.3        | 9.1      | 251               | 53             | 8
> Swin-Small      | 83.1        | 8.7      | 435               | 111           | 28
> Focal-Small     | 83.5        | 9.1      | 192                | 63             | 17
> ||
> ViT-Base/16    | 77.9        | 17.6     | 291             |  57          | 8
> Deit-Base/16  | 81.8        | 17.6     | 291             |   57         | 8
> PVT-Large       | 81.7        | 9.8       | 352              | 77            |  14
> ViL-Base          | 83.2        | 13.4     | 145              |  35           |  5
> Swin-Base       | 83.4        | 15.4     | 291              | 70            |  17
> Focal-Base      | 83.8        | 16.0     | 138              | 44            |  11
>
> In the above table, we do notice that Focal Transformers are slower than most of their counterparts except for ViL on low-resolution inputs. However, when the resolution increases, our models become more comparable and even faster than several methods. Particularly,  we notice that CvT and ViL become less efficient for high-resolution inputs. We suspect it is because CvT performs convolution and then the self-attention across the whole feature map, and ViL has several global tokens which attend to the whole feature map. In contrast, our Focal Transformer models are more flexible to different input resolutions, since it performs window-wise attention and has predetermined focal region size and focal window size that are independent of the input resolution. Likewise, Swin Transformers also have such property and thus can maintain good efficiency for higher resolution input.
>
> We further study which components in our Focal Transformers mainly cause the lower throughputs. Based on our model design, we suspect this is because of two reasons: 1) the overhead to extract the fine-grain and coarse-grain tokens at each transformer layer; 2) the extra computation after adding more keys and values. To get more understanding, we investigate the time cost for different ablated models. By measuring the speed for each of the ablated Focal-Tiny models in Figure 5 of our main submission, we obtain the numbers in the below table. To further factorize the time cost of key and value extraction and attention computation, we implement pseudo versions of Focal-Tiny-local and Focal-TIny-global which retain the same number of keys and values by appending pseudo tokens instead of extracting them from the (pooled) feature map. As we can see, local fine-grained local attention causes more computational cost than coarse-grained global attention. Moreover, extracting fine-grained local tokens introduces more overheads than extracting the coarse-grain global tokens. This inspires one of our future directions on how to extract the local tokens more efficiently.
>
> | Model                                    | Top-1 Acc | GFLOPs   | Throughput (imgs/s) |
> |---|---|---|---|
> Focal-Tiny                             | 82.2           | 4.92         | 319
> Focal-Tiny-local                    | 81.4           | 4.90         | 417
> Focal-Tiny-local-pseudo     | n/a             | 4.90         | 636
> Focal-Tiny-global                 | 81.6            | 4.59        | 516
> Focal-Tiny-global-pseudo   | n/a             | 4.59         | 633
> Focal-Tiny-window              | 80.1            | 4.49        | 731
>
> Based on the above comparisons, we notice the fine-grained local attention brings most of the extra time cost, and it is mainly due to the sub-optimal implementation of extracting surrounding fine-grained key and value tokens for each window. Currently, we implement this using rolling operations, which is two times faster than the conventional unfolding operation. However, we find this customized implementation is still under-optimized compared with an efficient cuda kernel as used in the convolution layer, especially for the earlier stage which has relatively higher feature resolution. We believe when an efficient cuda kernel implementation is available, the efficiency gap can be closed to the gap of FLOPs between different methods. Before that, we empirically studied how our models perform when removing a few time-consuming fine-grained local attentions. Specifically, we tried two variants, one is removing the fine-grained local attention at the first stage, the other one is removing local attention at all odd layers.  As we can see from the table below, these two variants have higher throughputs and slightly lower Top-1 accuracy, but still higher than the Swin-Transformer counterparts. These results imply that we may not need to have fine-grain local attention at all layers and all stages. We will have more systematic studies on this aspect.
>
> | Model                                                                      | Top-1 Acc | GFLOPs | Throughput (imgs/s) |
> |---|---|---|---|
> Focal-Tiny                                                               |     82.2       | 4.92        |  319
> Focal-Tiny (no local fine-grain at first stage)    |     82.1       | 4.77        | 388
> Focal-Tiny (local fine-grain at even layers)       |     81.9      | 4.75         | 399
> ||
> Focal-Small                                                               |    83.5       |  9.12       | 192
> Focal-Small (no local fine-grain at first stage)    |   83.5       |  8.98      | 217
> Focal-Small (local fine-grain at even layers)       |   83.3       |  8.85      | 240
> ||
> Focal-Base                                                            |    83.8       | 16.04    | 138
> Focal-Base (no local fine-grain at first stage)     |    83.7       | 15.80   | 154
> Focal-Base (local fine-grain at even layers)        |    83.5       |  15.56  | 172
>
> **Q2. Performance saturates for Focal-Base model.**
>
> A2: Thanks for pointing this out! During our submission, we also observed these close Top-1 accuracies for our Focal-Small and Focal-Base model. Based on our further experiments, we found this is due to the sub-optimal hyperparameter setting. More specifically, in our submission, we set the drop path rate 0.3 for our base model without heavy hyperparameter tuning. But when we increased it to 0.5 following Swin Transformer, we can get a maximal 83.8% (83.7% on average) Top-1 accuracy on ImageNet-1K. This suggests that when vision transformer models become larger and deeper, the drop path rate should also be increased accordingly for a good regularization.
>
> Based on the new drop path rate, we study whether we can further improve the performance of our Focal-Base model. One change we did is that we increase the focal region size from the original [7,5,3,1] to [9,7,3,1], which enables more global interactions at the first two stages. This change achieves a better average Top-1 accuracy 83.9%. When we further change the convolutional patch embedding layer from default non-overlapped (kernel size = stride size) to overlapped ones (kernel size > stride size), the performance can be further improved to 84.1% top-1 accuracy on average.
>
> All these ablation studies demonstrate that our Focal-Base model is not saturated when we use reasonable hyperparameters and can be further improved from various aspects.

---

### Official Review · Reviewer_B1T8 · 2021-07-16

**Rating:** 6
**Confidence:** 4

**Summary:**

This paper proposes a focal version of self-attention module for vision transformers. The query is kept at high resolution, but the number of keys is reduced depending on the relative position difference. This proposed attention is implemented within multi-scale windows. It shows good performance on top of recent transformer advances, on ImageNet and COCO.


**Limitations And Societal Impact:**

Yes.

**Main Review:**

## Strength
1. The proposed focal attention mechanism is original and novel.
2. The idea is clearly presented with equations and illustrations.
3. Training details included for fair comparison and reproducing. Comparison is fair with other methods.
4. Various settings and tasks are explored and compared with solid baselines.

## Weakness
1. Runtime / throughput is not clear. The proposed attention requires extra subsampling layers, key value computation, and more importantly, concatenating keys and values from different resolution windows (according to the code). These might introduce extra runtime cost with the same FLOPs. But this is not sufficiently discussed in the paper.
2. Scaling to larger models. The ImageNet performance of Focal-Base is the same as Focal-Small, and is similar to Swin with the same size. In addition, the improvement over Swin in the Small regime is limited as well. In this case, error bars may be reported to address result variances. However, Focal-Base improves consistently on detection segmentation tasks, given a limited ImageNet Top-1. Does it suggest that more context (larger focal size) is necessary for COCO than ImageNet? (Relates to the discussion in Line 281).
3. What is the intuition behind sharing the same set of query key parameter for both small and large windows? It does save parameters and FLOPs, but the model might be extracting local edges/corners with fine detail and global semantic concepts with coarse context windows. This seems to suggest that different (not shared) parameters might be good for fine and coarse levels. It might be worth ablating that three independent attention could be used for the three levels.
4. The training recipe seems modified slightly from DeiT/Swin, e.g. the drop path rate. Does it affect comparison fairness?

------------------------------------------------------------Post Rebuttal-------------------------------------------------------------------

After reading the author rebuttal and other reviews, I think the weakness 2 and 4 are mostly addressed. Nevertheless, the runtime/throughput is a confirmed weakness according to the authors' results in the rebuttal and is also a shared concern among reviewers. But I think this is ok given other contribution of the paper. In this case, I keep my original rating of 6.

**Time Spent Reviewing:**

3

---

> ### Author Response · Authors · 2021-08-10
> **Thanks for your valuable comments**
>
> We thank the reviewer for acknowledging the novelty and contributions of our work! We also thank the reviewer for all the valuable comments and answer the questions one by one below.
>
> **Q1: Runtime / throughput is not clear.**
>
> A1:  Thanks for pointing out this! Below we report the throughputs for our Focal Transformers with different model sizes. For comparison, we also report the throughputs for PVT, Swin Transformer, Vision Longformer, CvT, etc. All throughputs are measured on 1 V100 GPU with batch size 128 and input image resolution 224x224.
>
> | Model               | Top-1 Acc | GFLOPs | Throughput (imgs/s) |
> | -----------| ----------- | ----------- | ----------- |
> DeiT-Small/16    |      79.8       |  4.6     | 939
> PVT-Small         |      79.8       | 3.8      | 794
> CvT-13              |      81.6       | 4.5      | 746
> ViL-Small           |      82.0       | 5.1      | 397
> Swin-Tiny          |     81.2       | 4.5       | 760
> Focal-Tiny         |     82.2       | 4.9       | 319
> |  |
> PVT-Medium     | 81.2        | 6.7      | 517
> CvT-21              | 82.5        | 7.1      | 480
> ViL-Medium       | 83.3        | 9.1      | 251
> Swin-Small        | 83.1        | 8.7      | 435
> Focal-Small     | 83.5        | 9.1      | 192
> |  |
> ViT-Base/16    | 77.9        | 17.6     | 291
> Deit-Base/16  | 81.8        | 17.6     | 291
> PVT-Large       | 81.7        | 9.8       | 352
> ViL-Base          | 83.2        | 13.4     | 145
> Swin-Base       | 83.4        | 15.4     | 291
> Focal-Base      | 83.8        | 16.0     | 138
>
> In the above table, we do notice that Focal Transformers are slower than most of their counterparts except for ViL though they have similar FLOPs. We found this is because of 1) the overhead to extract the fine-grain and coarse-grain tokens at each transformer layer; 2) the extra computation after adding more keys and values. To get more understanding, we investigate the time cost for different ablated models. By measuring the speed for each of the ablated Focal-Tiny models in Figure 5 of our main submission, we obtain the numbers in the below table. To further factorize the time cost of key and value extraction and attention computation, we implement pseudo versions of Focal-Tiny-local and Focal-TIny-global which retain the same number of keys and values by appending pseudo tokens instead of extracting them from the (pooled) feature map. As we can see, local fine-grained local attention causes more computational cost than coarse-grained global attention. Moreover, extracting fine-grained local tokens introduces more overheads than extracting the coarse-grain global tokens. This inspires one of our future directions on how to extract the local tokens more efficiently.
>
> | Model     | Top-1 Acc | GFLOPs   | Throughput (imgs/s) |
> | -----------           | ----------- | ----------- | ----------- |
> Focal-Tiny                             | 82.2           | 4.92         | 319
> Focal-Tiny-local                    | 81.4           | 4.90         | 417
> Focal-Tiny-local-pseudo     | n/a             | 4.90         | 636
> Focal-Tiny-global                 | 81.6            | 4.59        | 516
> Focal-Tiny-global-pseudo   | n/a             | 4.59         | 633
> Focal-Tiny-window              | 80.1            | 4.49        | 731
>
> Based on the above analysis, the fine-grained local attention brings most of the extra time cost, and it is mainly due to the sub-optimal implementation of extracting surrounding fine-grained key and value tokens for each window. Currently, we implement this using rolling operation, which is two times faster than the conventional unfolding. However, we find this customized implementation is still under-optimized compared with a cuda kernel, especially for the high feature resolution at the first stage. We believe when an efficient cuda kernel implementation is available, the gap of time cost can be reduced to approach the gap of FLOPs. Before that, we empirically studied how our models perform when removing a few time-consuming fine-grained local attentions. Specifically, we tried two variants, one is removing the fine-grained local attention at the first stage, the other one is removing local attention at all odd layers.  As we can see from the table below, these two variants have higher throughputs and slightly lower Top-1 accuracy, but still higher than the Swin-Transformer counterparts. These results imply that we may not need to have fine-grain local attention at all layers. We will have more systematic studies on this aspect.
>
> | Model    | Top-1 Acc | GFLOPs | Throughput (imgs/s) |
> | -----------           | ----------- | ----------- | ----------- |
> Focal-Tiny      |     82.2       | 4.92        |  319
> Focal-Tiny (no local fine-grain at first stage)    |     82.1       | 4.77        | 388
> Focal-Tiny (local fine-grain at even layers)       |     81.9      | 4.75         | 399
> |  |
> Focal-Small                                                       |    83.5       |  9.12       | 192
> Focal-Small (no local fine-grain at first stage)    |   83.5       |  8.98      | 217
> Focal-Small (local fine-grain at even layers)       |   83.3       |  8.85      | 240
> |  |
> Focal-Base                                                         |    83.8       | 16.04    | 138
> Focal-Base (no local fine-grain at first stage)     |    83.7       | 15.80   | 154
> Focal-Base (local fine-grain at even layers)        |    83.5       |  15.56  | 172
>
> **Q2: Scaling to larger models.**
>
> A2: Thanks for pointing out this! During our submission, we also observed this. Based on our further experiments, we found this is due to the sub-optimal hyperparameter setting. More specifically, in our submission, we set the drop path rate 0.3 in one shot for our base model without any further hyperparameter tuning. But when we increased it to 0.5 as in Swin Transformer, we can get maximal 83.8% Top-1 accuracy on ImageNet-1K. This suggests that when vision transformer models become larger and deeper, the drop path rate should also be increased accordingly for a good regularization to prevent overfitting.
>
> For consistency to Swin models, we change the drop path rates for our Focal-Small and Focal-Base to 0.3 and 0.5, respectively, and conduct multiple rounds of training with all our three models as the reviewer suggested. We report the mean Top-1 acc and std below.
>
>
> | Model              |   Top-1 Mean  | Top-1 Std  |
> | -----------           | ----------- | ----------- |
> Focal-Tiny       |    82.15             |  0.105
> Focal-Small     |    83.51             |  0.055
> Focal-Base      |    83.69             | 0.1002
>
> Clearly, with the same hyperparameter settings, our Focal-Base has better performance than Focal-Small, and these three models consistently outperform the corresponding Swin Transformers.
>
> As the reviewer pointed out, our models indeed have more gains over Swin Transformers on COCO object detection than ImageNet classification when the model becomes larger and deeper. We suspect the reason why we observe this trend is because larger Swin Transformer models can already capture sufficient context through the window-shift mechanism for image classification. Consider the input resolution 224x224. From the first stage to the last stage, the feature map resolutions are 56x56, 28x28, 14x14, and 7x7, respectively. At the earlier stage, our Focal Transformers can capture more long-range dependencies than Swin Transformers. But when it comes to the later stages, the stacked window-shift operation in Swin Transformers can (almost) have a receptive field that covers the whole feature map, especially for larger models. However, for object detection, the input image resolution is usually much higher, e.g., 800x1333. In this case, the feature map at the last stage is still much larger than a window size 7x7. Even the base Swin Transformer is still difficult to capture the global context sufficiently. In contrast, our focal self-attention is intentionally designed to model such long-range dependencies for high-resolution feature maps. As such, we can still observe considerable improvements over Swin Transformers on COCO detection.
>
> **Q3: Intuition behind sharing qkv embedding layer for different focal levels.**
>
> A3: Thanks for this valuable suggestion! The reason for us to share the qkv embedding is two-fold: 1) though the tokens are from different granularities, we would like to project them into the same qkv space so that the attention between queries and keys from different granularities can be well-calibrated; 2) Indeed, using different qkv embedding layers will introduce more parameters. For example, using two qkv embedding layers for the local and global focal levels in our Focal-Tiny models will bring extra ~7M parameters on the original 29M model. We agree with the reviewer that different granularities may need different qkv embedding layers because they may capture different levels of structures. Due to the limited computational resource and time, we cannot report the performance for this new design right now, but we will post the results here once we get the experiments finished.
>
> **Q4: Slight difference on training recipe.**
>
> A4: Thanks for pointing this out! In our submission, we used drop path rates 0.2, 0.2, and 0.3 for our tiny, small, and base Focal Transformers, respectively. In contrast, Swin Transformers used 0.2, 0.3, and 0.5 drop path rates. We observed these differences indeed affect the performance to some extent. As shown in the above table, we reported the numbers with these new settings. Clearly, it helps us to achieve even better performance than our previous setting, especially for our Focal-Base model (83.5->83.8 maximally). Note that even with the sub-optimal hyperparameter setting, our Focal Transformers already outperform Swin Transformers on both image classification and object detection tasks.

---

> > ### Author Response · Authors · 2021-08-16
> > **Quick update with separate qkv embedding**
> >
> > We thank the reviewer again for the suggestion to use separate qkv embedding layers for different focal levels. Following this suggestion, we did a quick experiment to use two separate qkv embedding layers for the two focal levels. For Focal-Tiny model, this change brings extra ~7M parameters. In the meantime, it achieves 82.4% top-1 accuracy, which is 0.2% better than the original Focal-Tiny model. Based on this result, we agree with the reviewer that using different qkv embedding layers should be better to capture the different granularities of visual contents, but at the cost of extra model parameters. We will add this experiment to our revision.

---

### Official Review · Reviewer_rkny · 2021-07-20

**Rating:** 6
**Confidence:** 4

**Summary:**

This paper presents an efficient self-attention mechanism for vision transformer, which simultaneously capture both short and long-range visual dependencies by making the query to attend patches at different granularity levels. The authors demonstrate its performance on image classification and object detection.

**Limitations And Societal Impact:**

Yes.

**Main Review:**

Originality:
- The idea of extracting visual features using multiple granularity levels is well-established. Previous work has also tried to build a hierarchy of features within vision transformer (e.g., PVT). However, it is novel to build a single self-attention head which can attend multiple granularity levels.

Quality:
- The submission is technically sound.

Clarity:
- The paper is well written.

Significance:
- The empirical results are compelling.

Detailed comment:

1,  In section 3.1, a “patch embedding layer which consists of a convolutional layer with filter size and stride both equal to 4” could not transform input $\frac{H}{4} \times \frac{W}{4} \times (4\times4\times3)$ to output $\frac{H}{4} \times \frac{W}{4} \times d$.

2,  In Table 2, Focal-Base does not achieve better result than Focal-Small. Do you think it’s due to the limited set of hyperparameters that has been tried, or it’s due to the strong induction bias within focal attention may hurt the generalization ability of big model?

3, What is the criterion to organize Table 2? CvT-21 should be placed along with Focal-Tiny.

**Time Spent Reviewing:**

6

---

> ### Author Response · Authors · 2021-08-10
> **Thanks for your valuable comments**
>
> First of all, we thank the reviewer for acknowledging the novelty, quality, clarity of our work! We also thank the reviewer for all the valuable comments! We answer the questions one by one below.
>
> **Q1: Dimension inconsistency for convolutional patch embedding layer.**
>
> A1: Sorry for the confusion! By filter size equal to 4, we meant the kernel size for our convolution is set to 4x4. That being said, the convolutional layer we used for the first patch embedding is implemented with pytorch nn.Conv(in_channel=3, out_channel=d, kernel_size=4, stride=4). As such, the input image of size 3x224x224 will be transformed to dx56x56. Note that this convolutional layer is exactly the same as in Swin Transformer for a fair comparison.
>
> **Q2: Big model does not achieve better performance.**
>
> A2: Thanks for pointing this out! We also noticed this when working on our submission. Based on further experiments, we found this is due to the limited hyperparameter tuning, instead of the strong inductive bias or poor generalization ability of big models. More specifically, in our submission, we used a small drop path rate of 0.3 for our Focal-Base model. When we follow Swin Transformer to increase it to 0.5, we can get 83.8% Top-1 accuracy on ImageNet-1K. This indicates that when vision transformer models become larger, the drop path rate should also be increased accordingly for a good regularization on the big models.
>
> **Q3: Criterion to put CvT-21 in Table 2.**
>
> A3: Thanks for the suggestion! Though CvT-21 just has a slightly larger model size than our Focal-Tiny (32.0M v.s. 29.1M), we compare it with our Focal-Small for two reasons: 1) it has much higher FLOPs than Focal-Tiny (7.1 GFLOPs v.s. 4.9 GFLOPs), which means more computations are used in the model; 2) it has 21 transformer layers, which is comparable to 24 layers in Focal-Small but much more than 12 layers in Focal-Tiny. Admittedly, the comparison between CvT-21 and Focal-small is still not fair enough. To have a relatively fair comparison, we increase the depth of our Focal-Tiny model from 2-2-6-2 to 1-2-21-1 but decrease the hidden dimension from 96 to 64. These changes lead to a new version of Focal-Tiny model with 26.5M parameters and 4.67 GFLOPs. Based on this new Focal-Tiny model, we achieved 82.9% Top-1 accuracy, which is higher than 82.5 for CvT-21 but uses much fewer parameters and FLOPs. In our revision, we will carefully make fair comparisons with previous works including CvT by adding more variants of our Focal Transformer models to match the number of parameters and GFLOPs as possible as we can.

---

### Author Response · Authors · 2021-08-10
**We thank all reviewers for their valuable comments**

First of all, we thank all reviewers for their valuable comments! We are pleased all reviewers think our paper is well-written with a clear story. We are also encouraged that Reviewer rkny and B1T8 think our method is original and novel, and Reviewer HEkP thinks our design is reasonable to address an interesting research problem as acknowledged by Reviewer NEKg.

Based on all the constructive comments, we can summarize two common concerns about our work. Our separate responses to these two concerns raised by different reviewers may be slightly different based on the specific questions, but here we attempt to address the common pieces at the beginning to help align the discussion.

### 1. Concern about model efficiency/throughput

We thank the reviewers for raising this concern and suggesting us to report the throughputs. Indeed we found FLOPs cannot directly reflect the running speed of different methods. Below we report the throughputs for different methods, including Vision Transformer (ViT), DeiT, PvT, CvT, Vision-Longformer (ViL), Swin Transformers, and our own Focal Transformers. For comprehensiveness, we not only compare the throughputs with the conventional input resolution 224x224 for image classification but also compare on higher input resolutions 448x448 and 896x896 as suggested by Reviewer HEkP. We report the numbers below.

| Model | Top-1 Acc | GFLOPs | 224x224 | 448x448 | 896x896 |
| --- | --- | --- | --- | --- | --- |
DeiT-Small/16 | 79.8 | 4.6 | 939 | 101 | 20
PVT-Small | 79.8 | 3.8 | 794 | 172 | 31
CvT-13 | 81.6 | 4.5 | 746 | 125 | 14
ViL-Small | 82.0 | 5.1 | 397 | 87 | 17
Swin-Tiny | 81.2 | 4.5 | 760 | 189 | 48
Focal-Tiny | 82.2 | 4.9 | 319 | 105 | 27
||
PVT-Medium | 81.2 | 6.7 | 517 | 111 | 20
CvT-21 | 82.5 | 7.1 | 480 | 85 | 10
ViL-Medium | 83.3 | 9.1 | 251 | 53 | 8
Swin-Small | 83.1 | 8.7 | 435 | 111 | 28
Focal-Small | 83.5 | 9.1 | 192 | 63 | 17
| |
ViT-Base/16 | 77.9 | 17.6 | 291 | 57 | 8
Deit-Base/16 | 81.8 | 17.6 | 291 | 57 | 8
PVT-Large | 81.7 | 9.8 | 352 | 77 | 14
ViL-Base | 83.2 | 13.4 | 145 | 35 | 5
Swin-Base | 83.4 | 15.4 | 291 | 70 | 17
Focal-Base | 83.8 | 16.0 | 138 | 44 | 11
| |

In the above table, we notice that Focal Transformers are slower than most of their counterparts except for ViL on low-resolution inputs. When the resolution increases, our models become more comparable and even faster than several methods. Particularly, we notice that ViT/DeiT, CvT and ViL become less efficient for high-resolution inputs. We suspect it is because ViT/DeiT performs fined-grained global self-attention, CvT performs convolution, and then the global self-attention across the whole feature map. Though ViL performs local self-attention, it has several global tokens which attend to the whole feature map. In contrast, our Focal Transformer models are more flexible to different input resolutions, since it performs window-wise attention and has predetermined focal region size and focal window size that are independent of the input resolution. Likewise, Swin Transformers also have such property and thus can maintain very good efficiency for higher resolution input. Giving the above table, we refer the reviewers to our separate responses for a more detailed analysis. We will add all these comparisons and analyses into our revision. Above all, we would like to highlight here that the main motivation for us to propose the focal self-attention to model both short- and long-range interactions is that we would like to apply it to high-resolution inputs which are required for various dense prediction tasks, such as object detection, instance segmentation, and semantic segmentation.

### 2. Concern about Focal-Base model

Another main concern about our method is the seemingly saturated performance of our Focal-Base model. We also observed this when submitting our paper. Based on our follow-up experiments, we found this is not because our Focal Transformers is prone to overfit when the model becomes larger. Instead, it is mainly because of the sub-optimal hyperparameter setting, i.e., the drop path rates. Following Swin Transformer, when we changed the drop path rate 0.3 to 0.5 for our Focal-Base model, we obtained maximally 83.8% (83.7% on average over multiple runs) top-1 accuracy, which has a substantial margin to Swin-Base model. When we further increase the focal region size from [7,5,3,1] to [9,7,5,1] to capture more long-range dependencies for our Focal-Base model, we observed further improvement to 83.9% top-1 accuracy on average. These experiments clearly address the concerns raised by the reviewers.

### 3. Implementation details

Besides the above two main common concerns, some reviewers are confused by the implementation details of our Focal Transformer model. To resolve these concerns, we had replied separately and explained the computation process we used to implement the focal self-attention. More importantly, we will release the full code so that others can reproduce the results reported in our submission and here.

---

### Decision · Program_Chairs · 2021-09-27

**Decision:**

Accept (Spotlight)

**Comment:**

There is a unanimous agreement on the novelty/solid results of the paper and the AC agree with the reviewers. The proposed idea in this paper is interesting, well-motivated, and could make good contributions to the general vision and representation learning communities. In this regard, the AC recommends acceptance as spotlight.